# Dysmagnesemia in critically ill diarrheal patients in Bangladesh

**Aklima Alam[1], Gazi Md. Salahuddin Mamun ⬤[2]\*, Monira Sarmin[1], Shamsun Nahar Shaima[1,3], Gobinda Karmakar[1], Nurun Nahar Naila[1], Tahmeed Ahmed[1], Mohammod Jobayer Chisti[1]**

1  Nutrition Research Division, International Centre for Diarrhoeal Disease Research, Bangladesh (icddr,b), Dhaka, Bangladesh, 2  Infectious Diseases Division, International Centre for Diarrhoeal Disease Research, Bangladesh (icddr,b), Dhaka, Bangladesh, 3  Department of Nutritional Sciences, School of Graduate Studies, University of Toronto, Toronto, Ontario, Canada

\* gazi.mamun@icddrb.org

## Abstract

### Background

Despite having a pivotal role in numerous physiologic functions, magnesium disorders are rarely considered in clinical practice. This study aimed to explore the burden, predictors, and outcomes associated with magnesium imbalances among critically ill patients admitted to critical care settings with diarrheal disease.

### Methods

A retrospective chart analysis was done among critically ill patients with diarrhea aged more than 18 years admitted to the Intensive Care Unit (ICU) of a specialized hospital who had their serum magnesium measured. Data were extracted from an electronic health record system. Serum magnesium levels were measured upon ICU admission. Multivariate multinomial logistic regression analysis was done to find out the associations with clinical variables.

### Results

There was a higher incidence of hypomagnesemia (34%) than hypermagnesia (5.9%). On multivariate analysis, there were independent associations of hypomagnesemia with sepsis (mOR=6.25, 95% CI: 3.61 to 10.81, p<0.001), H/O regular medicine intake prior admission (mOR=1.94, 95% CI: 1.18 to 3.18, p=0.01). On the other hand, hypermagnesemia was independently associated with dehydration (mOR=4.78, p=0.003, 95% CI: 1.6 to 14.3). Comparing with other electrolyte disorders, hypocalcemia (p<0.001) was associated with hypomagnesemia. Hypermagnesemia was associated with hypochloremia (p=0.017), metabolic acidosis (p=0.014), and hypercalcemia (p=0.002).

**Data availability statement:** Based on the recommendation of its Ethical Review Board, the research administration of the International Centre for Diarrhoeal Disease Research, Bangladesh (icddr,b) has restricted making the personal information of hospitalized patients publicly available. However, data generated from icddr,b's electronic medical record can be provided to interested researchers for secondary data analyses upon approval of a data licensing application and agreement by the icddr,b data center committee. The data request may be sent to Ms. Shiblee Sayeed, Senior Manager, Research Administration, at shiblee_s@icddrb.org, who is responsible for addressing all data requests of icddr,b.

**Funding:** The author(s) received no specific funding for this work.

**Competing interests:** The authors have declared that no competing interests exist.

## Conclusion

The high occurrence of dysmagnesemia in our study highlights the need to closely monitor magnesium in critically ill ICU patients, particularly in resource limited settings. This could help prevent serious complications related to magnesium imbalances. Intensivists should remain alert to magnesium disturbances and conduct thorough patient evaluations.

---

## Introduction

Magnesium is a crucial intracellular cation, that plays a significant role in numerous physiological functions including maintaining electrolyte balance, stabilizing cell membranes, and regulating neuromuscular and cardiac functions [1]. Additionally, it supports the immune system by regulating inflammatory responses and alleviating oxidative stress [2].

Studies highlight the serious implications of low magnesium levels, linking them to doubled mortality risks and prolonged hospital stays among ICU patients [3]. Despite its significance, hypomagnesemia is often overlooked as a clinical risk factor, leading to magnesium being dubbed "the forgotten electrolyte" [4]. In contrast, hypermagnesemia is a relatively rare but serious electrolytic disorder, that arises in around 10–15% of hospitalized individuals affected by renal failure [5]. Untreated, hypermagnesemia can also be fatal, leading to cardiovascular problems like hypotension and arrhythmias, along with neurological symptoms such as confusion and lethargy in severe cases [6].

Diarrheal diseases are common in lower- and middle-income countries [7]. This often leads to fatal complications including severe sepsis, pneumonia, and urinary tract infections, significantly raising morbidity and mortality rates [8–10]. Acute watery diarrhea, often accompanied by dehydration and acute kidney injury, significantly disrupts fluid and electrolyte balances, potentially impacting body homeostasis [11,12].

Hypomagnesemia is a common but frequently overlooked electrolyte imbalance in critically ill patients, particularly those with diarrheal diseases, where gastrointestinal losses exacerbate deficiencies [13,14]. Emerging evidence highlights that magnesium depletion is often intertwined with other electrolyte disturbances, such as hypokalemia and hypocalcemia, due to its role in regulating renal potassium excretion and parathyroid hormone (PTH) activity [15,16]. For instance, hypomagnesemia impairs renal potassium conservation, perpetuating hypokalemia, while also inducing functional hypoparathyroidism, leading to refractory hypocalcemia [16,17]. These interdependent imbalances are especially consequential in ICU settings, where electrolyte dysregulation is linked to adverse outcomes like arrhythmias, prolonged mechanical ventilation, and increased mortality [18]. However, scant attention has been paid to understanding the prevalence and role of magnesium in diarrheal patients.

Importantly, patients with hypomagnesemia are more likely to develop sepsis, severe sepsis, and septic shock. This association stems from magnesium's role in immune function, endothelial integrity, and electrolyte balance [19,20].

Hypomagnesemia impairs neutrophil activity and phagocytosis, increasing infection susceptibility, while also exacerbating systemic inflammation and endothelial dysfunction [21].

This comprehensive study aims to explore the prevalence, predictors, and outcomes associated with magnesium imbalance among critically ill patients with diarrheal illnesses. By meticulously analyzing these imbalances in adults with diarrheal diseases, the research intends to bridge knowledge gaps, refine clinical management strategies, and enhance patient care, particularly in resource-constrained healthcare settings.

## Materials and Methods

### Study design

This retrospective study centered on adults aged 18 years or older admitted to the intensive care unit spanning from January 1st to December 31st, 2019. The primary investigation revolved around assessing their serum total magnesium levels. The defined thresholds for hypomagnesemia, normomagnesemia, and hypermagnesemia were serum magnesium levels below <0.65 mmol/L, within the range of 0.65–1.05 mmol/L, and above >1.05 mmol/L, respectively.

Study Site: The research was conducted at the Dhaka Hospital, which is part of the International Centre for Diarrhoeal Disease Research in Bangladesh (icddr,b). This hospital specializes in treating diarrheal diseases and serves a significant number of patients, attending more than 200,000 cases annually [22]. The hospital transitioned to a paperless system in 2009, managing all clinical and laboratory records through an electronic patient medical record system. Admission to the hospital was solely based on the presence of diarrhea, regardless of the presence or absence of related complications or additional health conditions. Evidence-based standard treatment protocols were strictly adhered to for managing dehydrating diarrhea, electrolyte imbalances, malnutrition, and pneumonia with different severity [23–25].

Population and Study setting: The study involved individuals aged ≥18 years who presented at the Dhaka Hospital with diarrhea. Within the hospital premises, there exists a nine-bed Intensive Care Unit (ICU) specifically designed to cater to critically ill patients. Patient management with critical care support at icddr,b Dhaka Hospital, used to adhere with evidence-based protocols for dehydrating diarrhea, electrolyte imbalances, sepsis, and severe pneumonia that include the management of adults with diarrhea-related complications. Patients with severe dehydration received cholera saline or normal saline in the Emergency Department, both of whom lack magnesium [26]. Serum magnesium levels were tested post-ICU admission, with no prior corrections for imbalances. For severe sepsis, intravenous resuscitation with isotonic fluids (Hartmann's solution or normal saline) was administered, with inotropes and vasopressors used in septic shock. Patients were monitored for adverse reactions, such as respiratory distress, facial flushing, restlessness, or altered mentation, until discharge.

### Data collection

A semi-structured case report form was designed and finalized to collect relevant study data from an electronic health record system called Sheba, at Dhaka Hospital of icddr,b. Patient anonymity was strictly maintained during the data extraction and analysis. Information gathered encompassed demographic details (age, gender) and clinical features upon admission, including diarrhea type, dehydration status, presence of features of pneumonia, abnormal mental status, hypoxemia, and sepsis, and symptoms like vomiting, fever, respiratory distress, and hypoxemia. Initial laboratory test results at admission, including serum levels of sodium, potassium, chloride, bicarbonate, total calcium, total magnesium, and creatinine, were recorded. Additionally, treatment history, such as antibiotic use during hospitalization, and outcome-related variables, like instances of hospital-acquired infections, need for ventilator support, and discharge status (including discharge, referral, leaving against medical advice, or death during hospitalization), were documented on paper forms [22].

## Working definitions

Diarrhea was characterized by the passage of three or more abnormally loose stools within a previous single day [27]. The assessment of dehydration utilized the Dhaka method which is almost identical to World Health Organization (WHO) method [28]. Fever was defined as an axillary temperature exceeding 38°C. Severe pneumonia and hypoxemia were classified by the WHO criteria [27]. The definitions and management of sepsis, severe sepsis, and septic shock were followed according to hospital protocol, improvised with further evidence from the surviving sepsis guidelines for diarrheal patients [10,29]. Our laboratory's defined range for serum magnesium, referred to as "Total Magnesium," ranged from 0.65–1.05 mmol/L. Hypomagnesemia was defined as a serum magnesium level below 0.65 mmol/L, and hypermagnesemia was defined as a level above 1.05 mmol/L [30]. Medicine means the history of regular medicine intake for chronic diseases like gastritis, bronchial asthma, diabetes, hypertension, etc. Acute kidney injury was identified when the serum creatinine level exceeded 1.5 times the standard age- and sex-specific upper limit for serum creatinine level [31].

## Data analysis

SPSS version 20 (IBM Corp, New York, USA) was used to input and STATA (version SE 15.0) analyzed the data. Descriptive statistics were used to explain clinical, socio-[32] demographic, laboratory, and other relevant data. Continuous variables were summarized using means with standard deviations for normally distributed data. Categorical data were summarized using frequency tables, and associations were analyzed using the chi-square test. To identify independent predictors of magnesium imbalances, multivariate multinomial logistic regression analysis was performed. Variables with a p-value <0.10 in the bivariate analysis, including age, gender, and comorbidities, were included in the final model (O'Brien and Fleming method) [33]. Adjusted odds ratios (aOR) with 95% confidence intervals (CI) were reported to quantify the strength of associations. Missing data were minimum (<5% in certain variables) and those were excluded systematically from the analysis.

## Ethics statement

Data were collected from the electronic medical records of children hospitalized in the intensive care unit. All information were anonymized and de-identified prior to analysis, allowing for a waiver of informed written consent. Additionally, a waiver for ethical approval regarding the disclosure of hospital data for this study was obtained from the Institutional Review Board of icddr,b.

## Results

In this study, 1,507 patients were admitted to the intensive care unit (ICU) during the study period. 987 of these individuals underwent serum magnesium testing. This research specifically targeted 388 (39%) eligible adults for serum magnesium analysis. Different serum magnesium levels were observed among the study patients: 132 (34%) showed hypomagnesemia, 233 (60%) had normal levels and 23 (5.9%) displayed hypermagnesemia (Fig 1).

### Normomagnesemia vs Hypomagnesemia

The adults with hypomagnesemia exhibited elevated body temperature, history of regular medicine intake, and a higher prevalence of sepsis, severe sepsis, septic shock, and increased levels of creatinine in comparison to individuals with normomagnesemia (Table 1).

Then, we conducted a multivariable multinomial logistic regression analysis, excluding inter-related variables to prevent clinical overfitting and ensure biological plausibility. After adjusting for potential confounders and considering significant variables from any bivariate analysis, our findings revealed independent associations of hypomagnesemia with sepsis and history of regular medicine intake for comorbidities (Table 2).

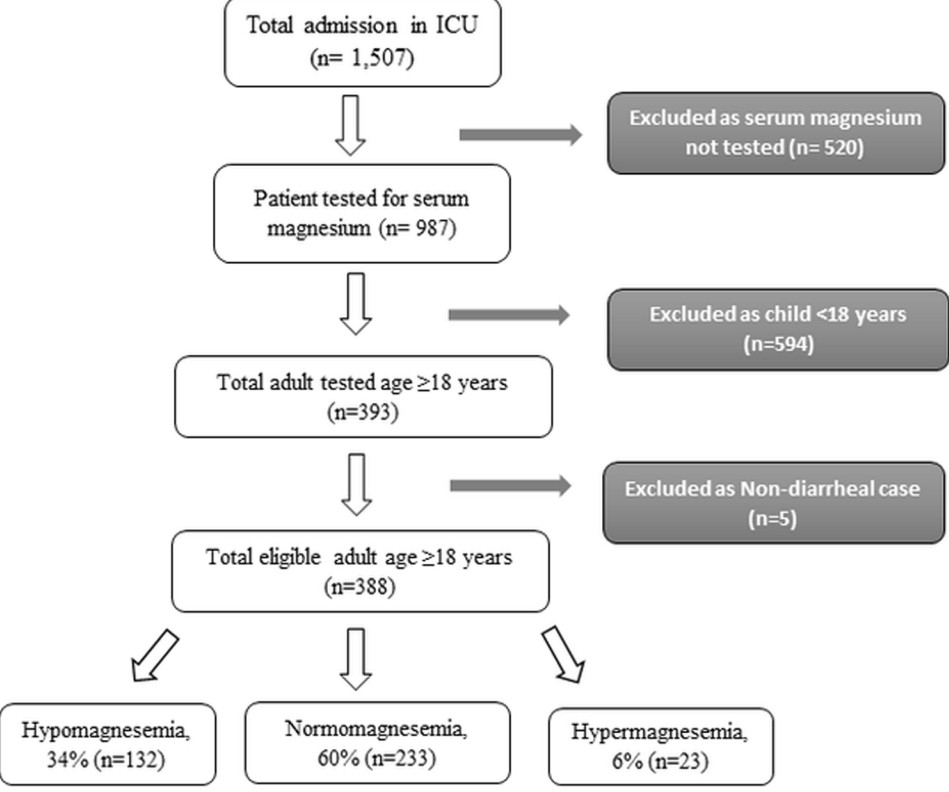

**Fig 1. Participant selection procedure.** This figure illustrates the steps involved in selecting participants for the study.

Unlike normomagnesemia, hypomagnesemia displays significantly lower levels of serum calcium. However, electrolytes such as sodium, potassium, chloride, and TCO2 exhibit nearly similar value levels in both normomagnesemia and hypomagnesemia (Table 3). No relationship has been discovered between normomagnesemia and hypomagnesemia concerning the disease's progression and outcomes (Supplementary Table 1).

### Normomagnesemia vs Hypermagnesemia

The adults with hypermagnesemia showed lower body temperature, elevated creatinine levels, and more prevalent dehydration compared to individuals with normomagnesemia (Table 4). Multinomial logistic regression indicated independent associations between hypermagnesemia and dehydration (Table 2).

In contrast to normomagnesemia, hypermagnesemia shows statistically significant association with hypochloremia, metabolic acidosis, and hypercalcemia (Table 3). There is no correlation found between normomagnesemia and hypermagnesemia regarding the disease's progression and outcomes (Supplementary Table 1).

### Discussion

To our knowledge, this is the first study to evaluate the prevalence, predictors, and outcomes associated with both hypomagnesemia and hypermagnesemia in critically ill diarrheal adults. Retrospective analysis of patient records found that hypomagnesemia was more prevalent than hypermagnesemia. The study also found independent associations of hypomagnesemia with sepsis, regular medicine use for comorbidities, and hypocalcemia. On the other hand,

**Table 1. The baseline clinical characteristics of adults with normomagnesemia and hypomagnesemia.**

| Variable | | Normomagnesemia (n = 132) | Hypomagnesemia (n = 233) | *p-value | 95% CI |
|---|---|---|---|---|---|
| | | n (%) | n (%) | | |
| Age (years) | Mean±SD | 49.47±21.17 | 48.65±17.25 | 0.689 | 1.00, 1.00 |
| | Range | 18-100 | 18-95 | | |
| Sex | Male | 64 (48.5) | 98 (42.1) | 0.236 | 0.5, 1.18 |
| | Female | 68 (51.5) | 135 (57.9) | | |
| Vomiting | | 92 (69.7) | 161 (69.1) | 0.905 | 0.61, 1.55 |
| Dehydration | | 56 (42.4) | 80 (34.3) | 0.125 | 0.46, 1.1 |
| Temperature (in°C), (mean±SD) | | 37.2±1.3 | 37.8±1.2 | 0.000 | 1.24, 1.8 |
| Medicine | | 51 (38.6) | 115 (49.4) | 0.049 | 1, 2.39 |
| Hypoxaemia | | 51 (38.6) | 112 (48.1) | 0.082 | 0.95, 2.27 |
| Mental status (altered) | | 48 (36.4) | 106 (45.5) | 0.090 | 0.94, 2.26 |
| Pneumonia | | 53 (40.2) | 117 (50.2) | 0.065 | 0.98, 2.32 |
| Bronchial Asthma | | 19 (14.4) | 42 (18.0) | 0.372 | 0.73, 2.36 |
| Cardiac disease | | 37 (28.0) | 65 (27.9) | 0.978 | 0.62, 1.6 |
| Diabetes Mellitus | | 15 (11.4) | 40 (17.2) | 0.139 | 0.86, 3.05 |
| Respiratory distress | | 70 (53.0) | 142 (60.9) | 0.141 | 0.9, 2.13 |
| Sepsis | | 60 (45.5) | 197 (84.6) | 0.000 | 4.01, 10.76 |
| Severe sepsis | | 36 (27.3) | 126 (54.1) | 0.000 | 1.98, 4.98 |
| Septic shock | | 18 (13.6) | 70 (30.0) | 0.001 | 1.54, 4.81 |
| AKI | | 92 (69.7) | 189 (81.1) | 0.004 | 1.26, 3.52 |
| Diarrhoea type | | | | | |
| Acute watery diarrhoea | | 112 (84.85) | 178 (76.39) | 0.064 (p-value) | |
| Invasive diarrhoea | | 20 (15.15) | 54 (23.18) | | |

This table summarizes the clinical features of adults with normal magnesium levels compared to those with hypomagnesemia upon ICU admission.

*CI: Confidence Interval, AKI: Acute Kidney Injury

hypermagnesemia was independently associated with dehydration. Additionally, hypermagnesemia was found to have significant hypochloremia and metabolic acidosis, and hypercalcemia.

Our study found that the prevalence of hypomagnesemia is 34% and hypermagnesemia is 5.9% in critically ill patients with diarrhea which is also similar to some studies among non-diarrheal cases. The prevalence of hypomagnesemia ranges from 25% to 59% [3,34–37] and hypermagnesemia ranges from 6% to 13.5% [34,35,38] in critically ill adult patients. Adults tend to experience higher rates of hypomagnesemia and lower rates of hypermagnesemia than children. In critically ill diarrheal children under 18 years, hypermagnesemia was present in 33.0% and hypomagnesemia in 5.2% [22]. Catalano et al also found hypomagnesemia rates were 9.52% in children under 18 years, 31.46% in those aged 19–65 years, and 59.01% in individuals over 65 years in Italy [37]. This variation between adults and children may be due to a potential reduction in magnesium absorption among the elderly [39].

Sepsis was also found to be independently associated with hypomagnesemia in our study. Similarly, a systematic review and meta-analysis confirmed this relationship by observing an independent association between the sepsis cascade and hypomagnesemia [40]. Immune dysregulations may contribute to the association between hypomagnesemia and increased recurrence of bacterial infections, including sepsis and severe sepsis [20,41]. Additionally, magnesium exhibits both endothelium-dependent and non-endothelium-dependent vasodilatory pathways in cases of advanced cascades of sepsis [42]. Studies found that magnesium can protect against sepsis cascades [43,44].

**Table 2. Association of Dysmagnesemia (either Hypomagnesemia or Hypermagnesemia) with normomagnesemia in Critically ill Diarrheal Adults.**

| Indicators | | Hypomagnesemia mOR (95% CI) | p-value | Hypermagnesemia mOR (95% CI) | p-value |
|---|---|---|---|---|---|
| **Dehydration** | | | | | |
| | *No* | Ref | | | |
| | *Yes* | 0.79 (0.48,1.3) | 0.360 | 4.78 (1.6,14.3) | 0.010 |
| **Mental status category** | | | | | |
| | *Normal* | Ref | | | |
| | *Altered* | 1.28 (0.79,2.09) | 0.310 | 1.13 (0.44,2.88) | 0.800 |
| **Sepsis** | | | | | |
| | *No* | Ref | | | |
| | *Yes* | 6.25 (3.61,10.81) | 0.000 | 2.3 (0.84,6.28) | 0.110 |
| **Medicine** | | | | | |
| | *No* | Ref | | | |
| | *Yes* | 1.94 (1.18,3.18) | 0.010 | 1.08 (0.42,2.79) | 0.870 |
| **Pneumonia** | | | | | |
| | *No* | Ref | | | |
| | *Yes* | 1.07 (0.66,1.75) | 0.770 | 0.46 (0.16,1.33) | 0.150 |
| **AKI** | | | | | |
| | *No* | Ref | | | |
| | *Yes* | 1.19 (0.66,2.14) | 0.560 | 6.34 (0.8,50.16) | 0.080 |

This table shows the statistical associations between dysmagnesemia and normal magnesium levels in ICU patients with diarrhea.

*mOR: multinomial odds ratio; CI: Confidence Interval; AKI: Acute Kidney Injury

**Table 3. The baseline biochemical characteristics of adults with normomagnesemia and Dysmagnesemia (either hypo or hypermagnesemia).**

| Variables | Normal n=132 | Hypomagnesemia n=233, 95% CI | p-value | Hypermagnesemia n=23, 95% CI | p-value |
|---|---|---|---|---|---|
| **Sodium** | 131.70±5.85 | 131.08±4.87(0.99, 1.01) | 0.466 | 130.54±3.67(0.99, 1.01) | 0.522 |
| **Potassium** | 4.33±1.11 | 4.06±1.01(0.99, 1.02) | 0.574 | 4.71±1.37(0.99, 1.02) | 0.911 |
| **Chlorine** | 100.83±6.79 | 100.69±6.12(0.99, 1.02) | 0.511 | 97.39±5.38(0.99, 1.02) | 0.017 |
| **TCO$_2$** | 16.75±4.87 | 16.59±4.32(0.99, 1.02) | 0.509 | 14.15±4.63(0.99, 1.02) | 0.014 |
| **Total Calcium** | 2.12±0.35 | 1.84±0.22(0.01, 0.06) | 0.000 | 2.39±0.46(0.01, 0.06) | 0.002 |

This table outlines the biochemical markers in patients with normal magnesium levels versus those with dysmagnesemia

* CI: Confidence Interval

In our study, regression analysis revealed an association between hypomagnesemia and regular medication usage. Additionally, previous research has demonstrated a link between prior use of medication for chronic disease and hypomagnesemia, which can occur through decreased gastrointestinal absorption or increased renal excretion of magnesium. [36,45]. Due to the majority of our patients either not providing their prescriptions or being unable to recall the names of their regular medications, we were unable to assess the relationship between the regular use of specific medicines and hypomagnesemia.

We found that adult patients with diarrhea and hypermagnesemia are more prone to dehydration due to the loss of fluids from diarrhea and/or vomiting, compared to those with normal magnesium levels. Furthermore, we found a significant association between hypermagnesemia and lower levels of chloride and bicarbonate, which has not been previously reported in the literature.

**Table 4. The baseline clinical characteristics of adults with normomagnesemia and hypermagnesemia.**

| Variables | | Normomagnesemia (n = 132) | Hypermagnesemia (n = 23) | *p-value | 95% CI |
|---|---|---|---|---|---|
| Age (years) | Mean±SD | 49.47±21.17 | 50.11±20.69 | 0.88 | 1.00, 1.00 |
| | Range | 18-100 | 20-102 | | |
| Sex | Male | 64 (48.5) | 12 (52.2) | 0.744 | 0.48, 2.81 |
| | Female | 68 (51.52) | 11 (47.83) | | |
| Vomiting | | 92 (69.7) | 19 (82.6) | 0.213 | 0.66, 6.46 |
| Dehydration | | 56 (42.4) | 18 (78.3) | 0.003 | 1.71, 13.95 |
| Temperature (in °C), (mean±SD) | | 37.2±1.3 | 36.5±0.9 | 0.011 | 0.35, 0.87 |
| Medicine | | 51 (38.6) | 9 (39.1) | 0.964 | 0.41, 2.53 |
| Hypoxaemia | | 51 (38.6) | 11 (47.8) | 0.408 | 0.6, 3.55 |
| Mental status (altered) | | 48 (36.4) | 10 (43.5) | 0.516 | 0.55, 3.3 |
| Pneumonia | | 53 (40.2) | 6 (26.1) | 0.205 | 0.19, 1.42 |
| Bronchial Asthma | | 19 (14.4) | 2 (8.7) | 0.466 | 0.12, 2.61 |
| Cardiac disease | | 37 (28.0) | 7 (30.4) | 0.813 | 0.43, 2.95 |
| Diabetes Mellitus | | 15 (11.4) | 2 (8.7) | 0.706 | 0.16, 3.49 |
| Respiratory distress | | 70 (53.0) | 14 (60.9) | 0.487 | 0.56, 3.4 |
| Sepsis | | 60 (45.5) | 14 (60.9) | 0.176 | 0.76, 4.61 |
| Severe sepsis | | 36 (27.3) | 9 (39.1) | 0.251 | 0.68, 4.31 |
| Septic shock | | 18 (13.6) | 5 (21.7) | 0.318 | 0.58, 5.33 |
| AKI | | 92 (69.7) | 22 (95.7) | 0.032 | 1.21, 71.62 |
| Diarrhoea type | | | | | |
| Acute watery diarrhoea | | 112 (84.85) | 21 (91.30) | 0.413 (p-value) | |
| Invasive diarrhoea | | 20 (15.15) | 2 (8.70) | | |

This table presents the clinical characteristics of adults with normal magnesium levels compared to those with hypermagnesemia.

*CI: Confidence Interval; AKI: Acute Kidney Injury

Our finding regarding the association of hypomagnesemia with hypocalcemia and hypermagnesemia with hyper-calcemia is understandable. There is abundant evidence indicating the frequent coexistence of hypocalcemia and hypomagnesemia [36,40,45,46]. This correlation arises from abnormalities in the synthesis and secretion of parathyroid hormone (PTH), along with the resistance of target organs to PTH. Additionally, magnesium deficiency can directly impact bone function, reducing calcium release regardless of PTH levels [35]. However, it was beyond our scope to evaluate this relationship between PTH, calcium, and magnesium as PTH was not tested for the study patients.

Additionaly, Hypomagnesemia significantly impacts ICU patient prognosis, increasing risks of arrhythmias, neurological dysfunction, and mortality [13,47]. Magnesium deficiency disrupts cardiac electrophysiology, predisposing patients to life-threatening arrhythmias like ventricular tachycardia and torsades de pointes [48]. Neurologically, it can cause seizures, delirium, and neuromuscular irritability, complicating recovery [49,50]. Early detection and correction of hypomagnesemia are critical, especially in high-risk patients with sepsis or heart failure, and routine monitoring with judicious supplementation should be integrated into ICU care to mitigate complications and optimize outcomes [34,51,52].

The findings of our study has the potential to have global implications. As hypomagnesemia is common in diarrheal patients who required ICU admission, our study findings may be generalizable in diarrheal adults requiring critical care support especially in LMIC settings.

## Limitations

The retrospective design and single-center nature among the diarrheal patients may limit the generalizability of our findings to broader populations. Importantly, we excluded missing data (<5%) from the analysis due to it`s retrospective nature. We did not assess changes in magnesium levels throughout the ICU stay, although during the study period, patients were monitored until their ICU discharge, potentially impacting the overall outcome parameters. Conducting a prospective study in a controlled setting could enhance the effectiveness of determining the participants' outcomes. Our study identified pre-admission medication use as a predictor of hypomagnesemia but lacked detailed data on specific drug classes known to influence magnesium homeostasis. Moreover, we couldn't analyze the nutritional data as this wasn't collected during their hospital stay.

## Conclusions

In conclusion, our study reveals a high incidence of hypomagnesemia among critically ill patients with diarrheal illnesses, with notable associations with sepsis, H/O regular medicine intake, and electrolyte disturbances. Hypermagnesemia was less common but associated with dehydration. These findings emphasize the need for careful monitoring and targeted interventions to manage magnesium levels in critical care settings, aiming to optimize patient outcomes.

## Declarations

## Supporting information

**Supplementary Table 1. Outcome between normomagnesemia and dysmagnesemia (hypomagnesemia or hypermagnesemia)**
(DOCX)

## Acknowledgments

We extend our deepest gratitude to our core donors, the Governments of Bangladesh and Canada for their unwavering support and commitment to icddr,b's research initiatives. We would also like to sincerely thank all clinical fellows, nurses, members of the feeding team, and hospital cleaners for their invaluable contributions to patient care.

## Author contributions

**Conceptualization:** Aklima Alam, Gazi Md. Salahuddin Mamun, Monira Sarmin, Tahmeed Ahmed, Mohammod Jobayer Chisti.

**Data curation:** Aklima Alam, Gazi Md. Salahuddin Mamun, Shamsun Nahar Shaima.

**Formal analysis:** Aklima Alam, Gazi Md. Salahuddin Mamun, Gobinda Karmakar.

**Methodology:** Gazi Md. Salahuddin Mamun.

**Supervision:** Gazi Md. Salahuddin Mamun, Mohammod Jobayer Chisti.

**Visualization:** Aklima Alam, Gazi Md. Salahuddin Mamun, Gobinda Karmakar.

**Writing – original draft:** Aklima Alam.

**Writing – review & editing:** Aklima Alam, Gazi Md. Salahuddin Mamun, Monira Sarmin, Shamsun Nahar Shaima, Gobinda Karmakar, Nurun Nahar Naila, Tahmeed Ahmed, Mohammod Jobayer Chisti.

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
