## [Decision Letter · Decision Letter 0]

11 Feb 2025

PONE-D-24-50871Dysmagnesemia in Critically Ill Diarrheal Patients in BangladeshPLOS ONE

Dear Dr. Mamun,

Thank you for submitting your manuscript to PLOS ONE. After careful consideration, we feel that it has merit but does not fully meet PLOS ONE’s publication criteria as it currently stands. Therefore, we invite you to submit a revised version of the manuscript that addresses the points raised during the review process.

We look forward to receiving your revised manuscript.

Kind regards,

Gaetano Paride Arcidiacono

Academic Editor

PLOS ONE

2. You have indicated that data is available from [shiblee_s@icddrb.org].  Please can we ask you to provide us with a general contact email address for the data requests, so readers can request access in perpetuity. If a general email is not available please provide a link to a website where readers can obtain access to data.

Additional Editor Comments:

The manuscript requires major revision, as suggested by the reviewers, before it can be considered for publication.

Reviewers' comments:

Reviewer's Responses to Questions

**Comments to the Author**

1. Is the manuscript technically sound, and do the data support the conclusions?

Reviewer #1: Yes

Reviewer #2: Partly

2. Has the statistical analysis been performed appropriately and rigorously? 

Reviewer #1: Yes

Reviewer #2: Yes

3. Have the authors made all data underlying the findings in their manuscript fully available?

Reviewer #1: Yes

Reviewer #2: Yes

4. Is the manuscript presented in an intelligible fashion and written in standard English?

Reviewer #1: Yes

Reviewer #2: Yes

5. Review Comments to the Author

Reviewer #1: Background; Adding literature that links magnesium with other electrolyte imbalances, such as hypokalemia or hypocalcemia, in patients with diarrheal diseases could provide a stronger rationale for the study's relevance in the ICU setting.

2. Materials and Methods; Clarifying how data were analyzed based on clinical variables such as age, gender, comorbidities, and nutritional status of patients, which may affect magnesium levels, would improve the method section.

3. Data Collection; Clarifying the time period for data collection and explaining whether any data were missing or incomplete would add transparency to the study methodology.

4. Operational Definitions; Adding the serum magnesium cutoff values used to define hypomagnesemia and hypermagnesemia will help readers understand the inclusion and exclusion criteria and facilitate a better understanding of the analysis.

5. Results; Including additional details about the clinical characteristics of the patients, such as age range, gender distribution, and comorbidities, would provide a clearer picture of the study population. Additionally, considering the different types of diarrhea (e.g., infectious vs. non-infectious) could provide more insight into magnesium imbalances.

6. Discussion; The discussion could benefit from a more detailed explanation of how hypomagnesemia impacts the prognosis of ICU patients and its potential complications, such as arrhythmias or neurological dysfunction. Additionally, highlighting practical implications for managing magnesium imbalances in critical care would be valuable.

7. Study Limitations; Acknowledging that the findings cannot be generalized to the broader population due to the retrospective design and the single-center setting is important. Additionally, explaining how missing data were handled in the analysis would strengthen the discussion of limitations.

Reviewer #2: The authors discuss magnesium imbalances (dysmagnesemia) in critically ill diarrheal patients in Bangladesh, examining their prevalence, predictors, and clinical impacts. While the study is of interest, the absence of an exploration into the causes of dysmagnesemia, the pathogens involved, and the management strategies applied significantly limits the relevance of the outcomes presented.

Please find attached my comments:

-It is noteworthy that patients with hypomagnesemia are more likely to develop sepsis, severe sepsis, and septic shock. Expanding on the mechanisms underlying this association in the introduction would provide valuable context.

-Although the study highlights medication intake prior to admission as a factor associated with hypomagnesemia, the lack of information on the types of medications limits the understanding of their potential role in magnesium imbalance.

-Since the study was conducted in a single hospital specialized in diarrheal diseases in Bangladesh, with inclusion criteria based solely on the presence of diarrhea (even in the absence of complications), the findings may not be generalizable to other populations or healthcare settings. It would be appropriate to further underline the causes of hospitalization and the necessity of hospital-level management.

-The study does not appear to have adequately controlled for factors such as nutritional status and the specific pathogens or diseases causing dysmagnesemia. These factors could significantly influence magnesium levels and clinical outcomes.

-The definition of acute kidney injury (AKI) used in the study does not follow the KDIGO guidelines (DOI: 10.1159/000339789). It would be more appropriate to adopt this standard. Furthermore, the cited reference for AKI pertains to infants, whereas the study focuses on adult patients.

-Similarly, the definition of septic shock would benefit from referencing current international guidelines. It would also be useful to specify how severe sepsis and septic shock were diagnosed. The current citation refers to a paediatric article, which may not be applicable.

-The lack of information on treatment and management strategies applied during hospitalization for dysmagnesemia limits the ability to evaluate the outcomes observed, thereby reducing the study's practical impact on clinical practice.

6. PLOS authors have the option to publish the peer review history of their article (what does this mean? ). If published, this will include your full peer review and any attached files.

**Do you want your identity to be public for this peer review?** For information about this choice, including consent withdrawal, please see our Privacy Policy .

Reviewer #1: No

Reviewer #2: No

---

## [Author Response · Author response to Decision Letter 1]

20 Mar 2025

Reviewer #1:

1. Background; Adding literature that links magnesium with other electrolyte imbalances, such as hypokalemia or hypocalcemia, in patients with diarrheal diseases could provide a stronger rationale for the study's relevance in the ICU setting.

Author Response: Thank you for reviewing our paper and your insightful suggestions that will definitely improve the quality of it. We have incorporated relevant literature discussing the association of magnesium with other electrolyte imbalances, such as hypokalemia and hypocalcemia, in patients with diarrheal diseases in ICU setting.

Hypomagnesemia is a common but frequently overlooked electrolyte imbalance in critically ill patients, particularly those with diarrheal diseases, where gastrointestinal losses exacerbate deficiencies. Emerging evidence highlights that magnesium depletion is often intertwined with other electrolyte disturbances, such as hypokalemia and hypocalcemia, due to its role in regulating renal potassium excretion and parathyroid hormone (PTH) activity. For instance, hypomagnesemia impairs renal potassium conservation, perpetuating hypokalemia, while also inducing functional hypoparathyroidism, leading to refractory hypocalcemia. These interdependent imbalances are especially consequential in ICU settings, where electrolyte dysregulation is linked to adverse outcomes like arrhythmias, prolonged mechanical ventilation, and increased mortality. [Pages 3–4, Lines 59–68 of track change version].

2. Materials and Methods; Clarifying how data were analyzed based on clinical variables such as age, gender, comorbidities, and nutritional status of patients, which may affect magnesium levels, would improve the method section.

Author Response: We thank the reviewer for their valuable suggestion. To address potential confounders, we performed a multivariate multinomial logistic regression analysis to identify independent predictors of magnesium imbalances. Variables with a p-value <0.10 in the bivariate analysis, including age, gender, and comorbidities, were included in the final model. This approach ensured that clinically relevant as well as biologically plausible factors were accounted for in the analysis. While nutritional status is an important factor, it was not included due to limitations in data availability. We have now explicitly described this methodology in the Statistical Analysis subsection to enhance transparency and rigor. [Page 8, Lines 148-158 of track change version].

3. Data Collection; Clarifying the time period for data collection and explaining whether any data were missing or incomplete would add transparency to the study methodology.

Author Response: We thank the reviewer for this important suggestion. The time period for data collection was from January 1st to December 31st, 2019. This is stated under methods section [Page 5, lines 84-85 of track change version]. Additionally, our analysis indicates that missing or incomplete data were minimal (<5%) for certain variables, ensuring the integrity of our findings [Page 7, lines 157-158 of track change version].

4. Operational Definitions; Adding the serum magnesium cutoff values used to define hypomagnesemia and hypermagnesemia will help readers understand the inclusion and exclusion criteria and facilitate a better understanding of the analysis.

Author Response: We appreciate the reviewer’s valuable suggestion. Hypomagnesemia was defined as a serum magnesium level below <0.65 mmol/L, and hypermagnesemia was defined as a level above >1.05 mmol/L. These cutoff values align with our institutional ISO certified laboratory reference value and are consistent with prior literature. These definitions have been added under the “Working definitions” section within the Methods [Page 7, Lines 135-137 of track change version].

5. Results; Including additional details about the clinical characteristics of the patients, such as age range, gender distribution, and comorbidities, would provide a clearer picture of the study population. Additionally, considering the different types of diarrhea (e.g., infectious vs. non-infectious) could provide more insight into magnesium imbalances.

Author Response: We thank the reviewer for their valuable suggestion. In response, we have now included additional details about the clinical characteristics of the study population. Specifically, we have added the age range and the gender distribution in the Results section. Also, we have reported percentages of comorbidities. Additionally, in LMIC settings differentiation of infectious from non-infectious diarrhea is often difficult due to a lack of routine performance of viral and bacterial pathogens (especially because of high cost of viral panel) causing diarrhea, thus we followed clinical classification of diarrhea (acute watery diarrhea and invasive diarrhea) provided by the WHO and we have added magnesium levels across these clinical types of diarrhea in Table 1 & Table 4.

6. Discussion: The discussion could benefit from a more detailed explanation of how hypomagnesemia impacts the prognosis of ICU patients and its potential complications, such as arrhythmias or neurological dysfunction. Additionally, highlighting practical implications for managing magnesium imbalances in critical care would be valuable.

Author Response: Thank you for your valuable suggestion. We have expanded the Discussion to provide a more detailed explanation of how hypomagnesemia affects the prognosis of ICU patients. Hypomagnesemia significantly impacts ICU patient prognosis, increasing risks of arrhythmias, neurological dysfunction, and mortality. Magnesium deficiency disrupts cardiac electrophysiology, predisposing patients to life-threatening arrhythmias like ventricular tachycardia and torsades de pointes. Neurologically, it can cause seizures, delirium, and neuromuscular irritability, complicating recovery. Early detection and correction of hypomagnesemia are critical, especially in high-risk patients with sepsis or heart failure, and routine monitoring with judicious supplementation should be integrated into ICU care to mitigate complications and optimize outcomes. [Page 19, Lines 284–291 of track change version].

7. Study Limitations; Acknowledging that the findings cannot be generalized to the broader population due to the retrospective design and the single-center setting is important. Additionally, explaining how missing data were handled in the analysis would strengthen the discussion of limitations.

Author Response: We appreciate the reviewer’s insightful comments regarding the limitations of our study. We acknowledge that the retrospective design and single-center nature among the diarrheal patients may limit the generalizability of our findings to broader populations. Missing or incomplete data were minimal (<5%) for certain variables, and we excluded those from the analysis, ensuring the integrity of our findings [Page 20, lines 297-300 of track change version].

Reviewer #2:

The authors discuss magnesium imbalances (dysmagnesemia) in critically ill diarrheal patients in Bangladesh, examining their prevalence, predictors, and clinical impacts. While the study is of interest, the absence of an exploration into the causes of dysmagnesemia, the pathogens involved, and the management strategies applied significantly limits the relevance of the outcomes presented.

Author Response: Thank you for your valuable time to review our manuscript. We’ve addressed your insightful comments as below in order to further improve the quality of our manuscript.

Please find attached my comments:

-It is noteworthy that patients with hypomagnesemia are more likely to develop sepsis, severe sepsis, and septic shock. Expanding on the mechanisms underlying this association in the introduction would provide valuable context.

Author Response: We thank the reviewer for this valuable suggestion. We have expanded the introduction to include potential mechanisms underlying the association between hypomagnesemia and sepsis.

This association stems from magnesium's role in immune function, endothelial integrity, and electrolyte balance. Hypomagnesemia impairs neutrophil activity and phagocytosis, increasing infection susceptibility, while also exacerbating systemic inflammation and endothelial dysfunction [Page 4, Lines 70-74 of track change version].

-Although the study highlights medication intake prior to admission as a factor associated with hypomagnesemia, the lack of information on the types of medications limits the understanding of their potential role in magnesium imbalance.

Author Response: We agree with your concern. Due to retrospective nature of this study the data on medication were non-specific, and thus we couldn’t collect the specific types of medications. We have highlighted this limitation in the revised manuscript and recommend future studes incorporate granular medication data to clarify their role [Page 20, Lines 305-307 of track change version].

-Since the study was conducted in a single hospital specialized in diarrheal diseases in Bangladesh, with inclusion criteria based solely on the presence of diarrhea (even in the absence of complications), the findings may not be generalizable to other populations or healthcare settings. It would be appropriate to further underline the causes of hospitalization and the necessity of hospital-level management.

Author Response: Thank you for your insightful suggestions. We acknowledge that the single-center design of our study, conducted in a hospital specializing in diarrheal diseases, may limit the generalizability of our findings to other populations and healthcare settings. This limitation has been stated in the Limitations section [Page 20, Lines 297-298 of track change version]. Additionally, we have provided further details on the causes of hospitalization and the hospital-level management protocols in the Methods section [Pages 5-6, Lines 101-111 of track change version].

-The study does not appear to have adequately controlled for factors such as nutritional status and the specific pathogens or diseases causing dysmagnesemia. These factors could significantly influence magnesium levels and clinical outcomes.

Author Response: We appreciate this important point. Nutritional status specific pathogens and causes were not analyzed due to absence of data in is retrospective cohort. We’ve now addressed this under the Limitations [page 20, lines 305-309 of track change version].

-The definition of acute kidney injury (AKI) used in the study does not follow the KDIGO guidelines (DOI: 10.1159/000339789). It would be more appropriate to adopt this standard. Furthermore, the cited reference for AKI pertains to infants, whereas the study focuses on adult patients.

Author Response: Thank you for your insightful comment. We’ve followed the definition of AKI according to KDIGO guideline where serum creatinine exceeding 1.5 times the standard age- and sex-specific upper limit is consistent with the KDIGO guidelines (DOI: 10.1159/000339789). [Pages 7, Lines 140-142, reference no. 30 of the track change version].

-Similarly, the definition of septic shock would benefit from referencing current international guidelines. It would also be useful to specify how severe sepsis and septic shock were diagnosed. The current citation refers to a paediatric article, which may not be applicable.

Author Response: Thank you for your comment. We’ve followed our hospital guideline for diagnosing and treating the cases of sepsis, severe sepsis and septic shock. We’ve now added this information under working definitions [page 7, lines 132-134 of track change version].

-The lack of information on treatment and management strategies applied during hospitalization for dysmagnesemia limits the ability to evaluate the outcomes observed, thereby reducing the study's practical impact on clinical practice.

Author Response: We agree that treatment strategies (e.g., magnesium supplementation) could influence outcomes. However, this observational study focused on prevalence and associations rather than interventions. Prospective studies evaluating magnesium correction protocols in this population are needed. We have added this as a future direction in the limitations (page 20, lines 304-305).

---

## [Editor Report · Decision Letter 1]

9 Apr 2025

Dysmagnesemia in Critically Ill Diarrheal Patients in Bangladesh

PONE-D-24-50871R1

Dear Dr. Mamun,

We’re pleased to inform you that your manuscript has been judged scientifically suitable for publication and will be formally accepted for publication once it meets all outstanding technical requirements.

Kind regards,

Gaetano Paride Arcidiacono

Academic Editor

PLOS ONE

---

## [Editor Report · Acceptance letter]

PONE-D-24-50871R1

PLOS ONE

Dear Dr. Mamun,

I'm pleased to inform you that your manuscript has been deemed suitable for publication in PLOS ONE. Congratulations! Your manuscript is now being handed over to our production team.

Kind regards,

on behalf of

Dr. Gaetano Paride Arcidiacono

Academic Editor

PLOS ONE